# Prioritization of the Target Population for Coronavirus Disease 2019 (COVID-19) Vaccination Program in Thailand

**DOI:** 10.3390/ijerph182010803

**Published:** 2021-10-14

**Authors:** Rapeepong Suphanchaimat, Titiporn Tuangratananon, Nattadhanai Rajatanavin, Mathudara Phaiyarom, Warisara Jaruwanno, Sonvanee Uansri

**Affiliations:** 1International Health Policy Program, Ministry of Public Health, Nonthaburi 11000, Thailand; rapeepong@ihpp.thaigov.net (R.S.); titiporn@ihpp.thaigov.net (T.T.); nattadhanai@ihpp.thaigov.net (N.R.); mathudara@ihpp.thaigov.net (M.P.); warisara.j@ihpp.thaigov.net (W.J.); 2Division of Epidemiology, Department of Disease Control, Ministry of Public Health, Nonthaburi 11000, Thailand; 3Bureau of Health Promotion, Department of Health, Ministry of Public Health, Nonthaburi 11000, Thailand

**Keywords:** migrant, COVID-19, vaccine, economic evaluation, cost-effectiveness analysis, cost-benefit analysis

## Abstract

Thailand was hit by the second wave of Coronavirus Disease 2019 (COVID-19) in a densely migrant-populated province (Samut Sakhon). COVID-19 vaccines were known to be effective; however, the supply was limited. Therefore, this study aimed to predict the effectiveness of Thailand’s COVID-19 vaccination strategy. We obtained most of the data from the Ministry of Public Health. Deterministic system dynamics and compartmental models were utilized. The reproduction number (R) between Thais and migrants was estimated at 1.25 and 2.5, respectively. Vaccine effectiveness (VE) to prevent infection was assumed at 50%. In Samut Sakhon, there were 500,000 resident Thais and 360,000 resident migrants. The contribution of migrants to the province’s gross domestic product was estimated at 20%. Different policy scenarios were analyzed. The migrant-centric vaccination policy scenario received the lowest incremental cost per one case or one death averted compared with the other scenarios. The Thai-centric policy scenario yielded an incremental cost of 27,191 Baht per one life saved, while the migrant-centric policy scenario produced a comparable incremental cost of 3782 Baht. Sensitivity analysis also demonstrated that the migrant-centric scenario presented the most cost-effective outcome even when VE diminished to 20%. A migrant-centric policy yielded the smallest volume of cumulative infections and deaths and was the most cost-effective scenario, independent of R and VE values. Further studies should address political feasibility and social acceptability of migrant vaccine prioritization.

## 1. Introduction

Coronavirus Disease 2019 (COVID-19), caused by Severe Acute Respiratory Syndrome Coronavirus-2 (SARS-CoV-2), was formally announced as a global health threat by the World Health Organization (WHO) on 30 January 2020 and has spread throughout the world [1]. Thailand was the first country outside of China to report a confirmed COVID-19 case. The COVID-19 pandemic in Thailand was under control until December 2020 through the application of numerous strategies including community-based contact tracing, isolation, quarantine, and border control [2]. The majority of clusters of cases were from superspreading events in nightclubs and a boxing stadium, which affected the majority of the Thai population [3]. The highest daily new cases numbered 188 in late March 2020 [4]. By mid-December 2020, the total case toll was 4246 [5].

The second wave began in late December and originated from the seafood market in Samut Sakhon, a province in the vicinity of Bangkok, which is a major residential area for migrant workers. The provincial economy relies significantly on fishery and industries that require a large migrant workforce [6]. The virus slipped through the Thai–Myanmar border via undocumented migrants, confirmed by genome sequencing [7]. Compared with the first wave in 2020, the number of cases was exceedingly high, with 1548 positive cases within the first week of the outbreak. Plausible explanations were poor work hygiene, crowded accommodation, and limited access to testing among migrant workers [8].

In early 2021, the global discourse and campaigns for COVID-19 mass vaccination caught the wider public’s attention. It is believed that the vaccine is the ultimate weapon to fight COVID-19 and curb outbreaks all over the world, leading to next steps in fighting the pandemic beyond non-pharmaceutical interventions (NPI) [9].

With the belief that vaccination could be an endgame for the COVID-19 pandemic and amid the backdrop of the surge of COVID-19 cases in Thailand, there arose an important policy question. Is it effective to implement a vaccination policy in the center of the epidemic outbreak, in this case, Samut Sakhon? More importantly, how can target populations for vaccination be prioritized, given the limited supply of vaccines? It was estimated that the first batch of COVID-19 vaccines would arrive in Thailand in the first quarter of 2021. The Thai government purchased live-attenuated COVID-19 vaccines from China for emergency relief [10] and planned to use viral-vector vaccines produced domestically as the main vaccine for later months [11].

This study therefore aimed to identify the effectiveness of diverse vaccination strategies in terms of the benefit of reducing the case toll, the reduction of mortality, and the resumption of economic activity from preventing cases and deaths, compared with the vaccine investment cost in light of the limited supply of vaccines. We hope that the findings from this study will help aid policy decisions to identify an optimal vaccination strategy that most benefits society as a whole.

## 2. Materials and Methods

### 2.1. Study Design

The research team mainly undertook secondary-data analysis. The parameters used in the model were obtained from two strands. First, we conducted a document review of the internal database of the Department of Disease Control (DDC) and relevant official authorities, such as the National Health Security Office (NHSO) and the National Statistical Office (NSO). Some key parameters, such as the incubation period and infectious duration, were retrieved from international literature. Second, we held a series of consultative meetings with public health experts and epidemiologists in the DDC during January–February 2021. The expert panel provided advice for the research team to construct the model and suggested some key parameters given a lack of primary evidence. More details on the parameters are explained in later subsections.

### 2.2. Model Framework

We used a compartmental susceptible-exposed-infectious-recovered (SEIR) model combined with a system dynamics (SD) model as a base framework [12]. The simplified framework is demonstrated in Figure 1.

The model categorized people into the susceptible, the exposed, the infectious, and the recovered. The rate of moving from being susceptible to being exposed was influenced by the reproduction number (R) [13]. The speed of transferring from the exposed category to the infectious category was determined by the incubation period. One of the key differences between our model and the traditional SEIR model is that we divided the infectious category into two subgroups, before isolation and after isolation, since in real practice, we did not let an infectee interact with susceptible people throughout the whole infectious period. Once admitted in a hospital, an infectee was considered isolated (supposing no nosocomial infection). The length of stay (LOS) in a hospital governed how fast a patient transferred from the infected category to the recovered category. As LOS was influenced by clinical severity, we further divided the population into four categories: asymptomatic, symptomatic but not needing intensive care (mild-to-moderate), symptomatic needing intensive care (severe), and dead. Another key difference from the conventional SEIR model was that we split the population into three groups: (i) Thais, (ii) registered migrants, and (iii) unregistered migrants (sometimes known as undocumented migrants). The rationale of this approach was that the assignment of authorities responsible for the cost of vaccination depended on who was to be vaccinated. For Thais and registered migrants, vaccination costs were shouldered by the Thai government. For unregistered migrants, there was a policy discourse proposing that vaccination costs should be covered by the private sector or entrepreneurs.

We classified each subpopulation into two strands: the vaccinated and the non-vaccinated. We supposed the only population to be vaccinated came from the susceptible category, and the vaccination rate was governed by the vaccination operation time set by the government. At the time of writing, the research on the effectiveness of vaccines from various companies had not been finalized. Therefore, we referred to the recommendation of the WHO, suggesting that a widely deployed COVID-19 vaccine should have a protective effect against infection of at least 50% (based on a placebo-controlled trial). With this respect, we used a figure of 50% as a vaccine effectiveness parameter for reducing the probability of switching from the susceptible category to the exposed category [14].

### 2.3. Model Assumptions

The analysis relied on the following assumptions.

#### 2.3.1. Assumptions Regarding the Population Components

First, we used hypothetical demographic figures that mimicked the demography of Samut Sakhon. The volume of Thai residents in Samut Sakhon from the civil registry was about 500,000 and the number of migrant workers registered with the Ministry of Labor (MOL) was approximately 270,000 [15,16]. We assumed that the total of unregistered migrants was equal to one-third of registered migrants [17]. Thus, the figure of 90,000 was used to reflect the volume of unregistered migrants.

Second, as the second wave of the outbreak heavily involved migrants, we postulated that R varied between Thais and migrants. For Thais, the value of nationwide R as monitored by the DDC at that time was about 1.25. This figure was validated against actual reported infectees. For migrants, at the beginning of the outbreak, R rose to 4, then gradually subsided. We hypothesized that if the outbreak still persisted, the migrant-specific R would maintain its high by doubling the value of R among Thais. Therefore, the value of 2.5 was applied to reflect the R among migrants.

Third, since in reality the volume of initial infectees could not be exactly known (because when the outbreak was detected, 2–3 generations of infectees had already passed by), we acquired this number from model calibration (by determining the number of initial infectees which could yield the daily incident cases from model estimation closest to the actual daily reported cases). The expert panel also suggested that the percentage of initial infectees among migrants should be much greater than Thais by a factor of 10:1. Based on all the above accounts, we used the figures of 2% and 0.2% as the percentage of initial infectees for migrants and Thais, respectively.

Fourth, we postulated that the probability of contact depended on the perception of new cases in the past 14 days. If the number of new daily cases was reduced by half, the contact rate would increase by a maximum of 50% among the vaccinees and 25% among non-vaccinees. More details of the calculation for this assumption are displayed in Appendix A.

Fifth, during 2020–2021, the Thai government promulgated various non-pharmaceutical interventions (such as compulsory face-mask policy in public spaces and physical distancing measures) with an aim to topple the outbreak [18]. This meant that, even without vaccination, the country had been exposed to default policies that mitigated the progression of the outbreak. Previous domestic literature suggested that the default policies contributed to a 50% effectiveness in protecting against COVID-19 in the population [19].

Last, we attempted to account for both intra-group and inter-group contacts, as, in practice, the chance of contact was not homogenous even within the same province. In other words, most Thais dwelled in Thai communities, and most migrants lived in migrant communities. We presumed that 60% of contacts among Thais happened within Thai communities, and the remaining 40% chance was equally shared between registered and unregistered migrants. The same idea was applied for migrants, as presented in Table 1.

#### 2.3.2. Assumptions Regarding the Parameter Values

First, we hypothesized that vaccination effectiveness would reach its peak within 30 days, then be sustained for 180 days, and subside to null by day 365.

Second, the 50% vaccine effectiveness implied not only the reduction of COVID-19 infection by half, but also alteration in clinical severity. In other words, we presumed that COVID-19 vaccines reduced the probability of facing severe clinical conditions or death by 50% (though we were aware that most literature on COVID-19 reported almost 100% effectiveness in preventing death, in this study, we used a much humbler approach; therefore, the 50% figure was used instead) [20].

Third, in line with the national healthcare policy of the government, the cost of treatment for COVID-19 patients (regardless of nationality status) was covered by the state, as COVID-19 was listed in the list of national dangerous communicable diseases in the “Communicable Diseases Act B.E.2558 (2015)” [18]. However, the cost of vaccination was another issue. The vaccination service for Thais and registered migrants was free of charge (as it was part of the insurance benefit) but not free for unregistered migrants. The vaccination cost for unregistered migrants was borne by the employers of these migrants.

Last, we needed to estimate the economic contribution of migrants to the gross domestic product (GDP) of Samut Sakhon. Prior research estimated that migrant workers all over the country contributed to 4.3–6.6% of the Thai GDP [21]. Since the proportion of migrants in Samut Sakhon’s population was at least three times greater than the corresponding proportion for the whole country, we therefore postulated that the economic contribution of migrants within Samut Sakhon was up to 20%.

### 2.4. Model Scenarios and Outcomes of Interest

At the time of writing, the policy in practice planned to vaccinate at least 200,000 people (equivalent to 400,000 doses) in Samut Sakhon. The government also delivered a strong policy direction that the prioritization of vaccines should cover at least two subpopulations: (i) Thai healthcare workers and those living in the epidemic center (so-called “special-risk”), and (ii) Thai nationals with underlying chronic diseases and elders aged above 60 (so-called “high-risk”). The big question was how to allocate COVID-19 vaccines to the rest of the population. Therefore, the expert panel proposed four policy scenarios (A–D). Policy A intended to allocate the rest of the vaccines to low-risk Thais (those who were neither identified as special-risk nor high-risk). Policy B planned to deliver the vaccines to registered high-risk migrants first, then the rest to low-risk Thais. Policy C was considered an expansion of policy B, this time extending vaccination coverage to both registered and unregistered high-risk migrants. Then, the rest of the vaccines were to be administered to low-risk Thais. Policy D allocated the vaccines to (both registered and unregistered) high-risk migrants first, then allotted the rest to low-risk registered migrants. Of note is that the risk distribution between Thais and migrants varied. The percentage of high-risk Thais was 15%, while the same percentage of migrants was only 5% (as may more migrants were of working age). We assumed the same figure (1%) for migrants and Thais facing special risk. Table 2 displays the risk distribution of the target populations.

To facilitate communication, we then named each policy option as follows: Policy A = Thai-centric, Policy B = High-risk-centric, Policy C = High-risk-centric plus, and Policy D = Migrant-centric. Table 3 presents details on the policy options of interest.

The outcomes of interest were divided into three parts: (i) case outcomes, (ii) cost outcomes, and (iii) cost-effectiveness outcomes. For the first part, we focused on the following indicators: daily incident cases, cumulative cases, prevalence of intensive care unit (ICU) beds, and cumulative deaths. The ICU beds were occupied by severe and dead cases. For the second part, we targeted the cost borne by public facilities (in other words, focusing on provider perspective) and GDP losses. For the third part, we used a no-vaccination scenario as a reference. Then, we assessed the outcomes in the following comparison pairs: (i) policy A vs. no vaccination, (ii) policy B vs. no vaccination, (iii) policy C vs. no vaccination, and (iv) policy D vs. no vaccination. We then assessed cost per case averted, cost per death averted, and cost per one Baht GDP resumed.

### 2.5. Parameter List and Formula

We used Stella 2.0 (number: 251-401-786-859) to run the model. Table 4 and Table 5 display essential parameters and formula used for the calculation.

### 2.6. Sensitivity Analysis

We also evaluated the change in the outcomes by varying the values of the reproduction number and vaccine effectiveness. The analysis was conducted in two steps. First, we fixed R at the values of 1.25 (for Thais) and 2.5 (for migrants) but altered vaccine effectiveness against infection to 20% and 80%. Second, we fixed vaccine effectiveness at 50% but serially altered R for migrants to 1.5, 2, and 3.

## 3. Results

We first present the number of cases in Figure 2. The trend and pattern of the daily incident cases in all vaccination scenarios were similar to each other, except the case magnitude. The no-vaccination policy demonstrated the largest daily case toll compared with other policy scenarios. The peak of the daily case toll reached over 1500 around days 1–3 and again at days 28–30, with multiple small peaks occurring throughout the entire study period. The peak varied by about 600–1500 cases during days 1–30, then subsided to below 600 after approximately day 60. Policy D showed the lowest daily case toll relative to all other policies. After day 60, the magnitude of the cases was approximately half of the peak case toll of the no-vaccination policy.

The volume of cumulative cases by different policy options is displayed in Figure 3. During the first 30 days, the cumulative case toll appeared to be approximately similar in all policy scenarios. The gap across policy options became obvious after day 60. The no-vaccination policy demonstrated a cumulative case toll of about 70,000 by the end of the year. Policy D displayed the lowest case toll of about 50,000, whereas other policy options yielded a case toll of about 60,000–65,000. After day 100, the total case volume for all scenarios appeared to reach a plateau with a very small growth rate per day (Figure 3).

Policy D also resulted in the lowest demand for ICU beds needed. The peak prevalence of ICU-bed demand amounted to about 600 by day 30. Apart from the no-vaccination policy, policy A caused the largest ICU-bed demand at approximately around 700 beds. In addition, the ICU bed demand for policy D started to subside by approximately day 45 and onwards, while the Thai-centric policy demand declined after day 55. All policy options met a plateau from day 180 and onwards (Figure 4).

In terms of deaths, the cumulative death toll at the end of the year was about half the toll in the non-vaccination policy scenario (365 deaths vs. 651 deaths). The second- and third-smallest death tolls were presented in policy scenarios C and B, respectively. Policy A also exhibited a relatively large death toll (about 550), the largest figure among all policy scenarios if the no-vaccination policy is set aside (see Figure 5).

The state appeared to invest most in policy A, relative to other policy options. The no-vaccination policy was the second-most cost-saving policy, after policy D. By the end of the year, all policy options except policy D contributed to almost the same total cost of about 2000 million Baht. Policy D entailed the least expense, at approximately 1600 million Baht. Note that before day 100, the no-vaccination policy was the greatest cost-saving option, but after that day, its cost outstripped the cost of policy D. All policies reached a plateau by approximately day 150 (Figure 6).

All vaccination policies caused a reduction in GDP loss compared with a no-vaccination policy. The greatest GDP saving was found in policy D, with approximately a 30% reduction in GDP loss relative to a no-vaccination policy (650 million Baht vs. 900 million Baht). Other policy alternatives meant a GDP loss of approximately 800 million Baht (Figure 7).

With reference to a no-vaccination policy, policy D appeared to be the best cost-saving option. The incremental cost to save a life, prevent further cases, or resume one Baht of GDP includes a minus sign, denoting less cost invested with some benefits gained. Apart from policy D, policy C seemed to be the most cost-effective alternative (for instance, it cost 153,940 Baht per one life saved, almost ten times less costly than policy A). The incremental cost per one Baht GDP resumed in Policy A was expected to be 1.7 Baht, about double the corresponding cost for policy B and about eight-fold larger than policy C (see Table 6).

The sensitivity analysis demonstrated that, when varying the reproduction numbers among migrants, policy D remained the most cost-effective, compared with other policy scenarios. We also noticed that the greater the R value, the more cost-effective this was. For example, given a reproduction number among migrants equal to 2, the incremental cost per one Baht GDP resumed for policy D (with reference to no-vaccination) was 2.2, but this ratio was enlarged greatly to 14.2 when the reproduction number equaled 1.5 (see Table 7).

By varying VE (but fixing the reproduction number), it is clear that the greater the VE, the more cost-effective the policy options. Under the assumption that VE equaled 80%, policies B, C, and D turned out to be a cost-saving option (as evidenced by the minus sign in the incremental cost per one Baht GDP resumed). In contrast, if the VE diminished to 20%, all policy options became less cost-effective. For instance, the incremental cost per death prevented by policy A (relative to no-vaccination policy) was enlarged from about 1.4 million to 3.6 million Baht. Policy D remained the most cost-effective option given the change in VE (see Table 8).

## 4. Discussion

This research is probably among the first of studies on the cost-effectiveness of a COVID-19 vaccination program that includes migrants and investigates further the issue of prioritization. Overall, our findings revealed that different policy options led to different public health outcomes. The migrant-centric vaccination strategy appeared to be the most effective option, both when considering cost per life saved and cost per macro-economic activity resumed. This message is very critical when considering a vaccination plan; that is, the vaccination policy in Thailand should not follow a nationalist approach by vaccinating only Thais while ignoring non-Thais (most of whom were migrants) on Thai soil. Moreover, this paper highlights that the policy should not only include migrants in the vaccination scheme but also identify migrants as a priority for vaccination in order to maximize the public health and economic benefits for the whole of society. A prior study in Samut Sakhon [27] suggested that in settings where physical distancing was difficult to exercise, vaccination measures should be urgently implemented. The International Organization for Migration has called on governments around the world to count and include all migrants in their territories (regardless of their legal immigration status) in national COVID-19 vaccine distribution plans [28,29].

A likely explanation for this is that the vaccination strategy yields the best return if it targets the population that mainly influenced the force of infection (the large reproduction number). In this study, we hypothesized that the migrant population faced a larger reproduction number than Thais. This assumption was in line with the fact that most migrants live in crowded conditions that accelerate disease transmission. Many countries have applied the same vaccination strategy in mitigating outbreaks; that is, prioritizing vaccine administration to the populations likely to cause super-spreading events. The experience of Singapore in dealing with COVID-19 outbreaks among migrants clearly exemplified this point. Singapore’s worst outbreak of COVID-19 occurred in the dorms of migrant workers in early 2020. It took several months to control the outbreaks. When vaccines arrived in Singapore, the Singaporean government implemented a policy to expedite the vaccination for migrant workers who had not been infected. As of June 2021, a fifth of migrants in the dorms were fully vaccinated against COVID-19 [30].

Studies on vaccine prioritization are quite sparse compared with other kinds of studies on COVID-19. In addition, the models used in vaccine prioritization studies always differ in details (assumptions, structures, and parameters, for instance), meaning that a direct contrast of outcomes across studies is impossible [31,32]. Matrajt et al. [33] suggested that optimal strategies to minimize deaths would either exclusively target groups with high fatality rates (high-risk populations) or would target groups with high rates of infection (for instance, migrant adults). Buckner et al. [32] proposed a “dynamic prioritization” strategy as the most effective means to minimize deaths and years of life lost. The dynamic prioritization strategy was a vaccination plan that first targeted people with a high risk of infection and then switched to targeting groups with high fatality rates. Buckner et al. [32] flagged that the VE against transmission was not always perfect (Buckner et al. called this a leaky vaccine). Thus, the dynamic prioritization strategy would likely yield the most favorable public health outcomes. Bubar et al. [31] proposed a slightly different vaccination strategy by highlighting that a highly effective transmission-blocking vaccine should be prioritized for adults aged between 20 and 49 years to minimize cumulative incidence, but mortality and years of life lost would be minimized if the vaccines were allocated to adults greater than 60 years of age first.

Since, currently, most of the research (despite being scarce) on COVID-19 vaccines has focused on public health outcomes (such as case aversion, death aversion, and years of life saved) or intent to receive the vaccines [34,35], we conducted further analysis by including macro-economic indicators (in this case, GDP recovery). From another angle, some studies [36,37,38] explored the macro-economic consequences following COVID-19, but those studies were delinked with vaccine prioritization. In addition, our sensitivity analysis found that even with a very humble assumption of VE (VE = 50% for base analysis and VE = 20% and 80% for sensitivity analysis), the benefit of mass vaccination, especially on outbreak-prone communities such as migrant populations, was still evident. Our findings pointed in the same direction as the study by Wang et al. [39], which applied the Markov decision tree for cost–benefit analysis on COVID-19 vaccines. However, the vaccines of interest in Wang et al.’s study were mRNA or viral-vector vaccines, which demonstrated a higher degree of VE (70–75%) relative to live-attenuated vaccines such as those in our study.

Despite a thorough analysis, some limitations remained in this study. The first is the issue of generalizability of the study findings. This is because the model primarily aimed to assist the decision-making of the government concerning the epidemic crisis in Samut Sakhon or similar areas where migrants are hugely populated. Second, during the crisis, it is extremely difficult to assess real-time empirical data. Therefore, for some parameters, we inevitably relied on a secondary data source or even opinions of the experts. Greater availability of empirical and more comprehensive data would allow a more complex analysis or an analysis that better reflects real-world population dynamics, such as stochastic modelling or agent-based modelling. Third, our analysis did not capture all aspects of policy implementation. Hence, in the actual setting, real-world outcomes might not be consistent with our findings, as there were many other uncontrollable or even unobserved factors. Such factors included logistic hurdles, delay of vaccine supplies, budget burden, vaccine hesitancy (though this issue might not be critical as, so far, based on a public poll, almost 90% of Thai residents were willing to be vaccinated [40]), social acceptability, and misperception of vaccine information. False information via online communication was ubiquitous during the time of crisis. In many countries, though, the authorities deployed several surveillance tools to slow down the rapid spread of misinformation online [41]. The societal belief that vaccines would be mainly distributed to migrants or people with precarious status first might compromise the policy decision to vaccinate migrants, despite evidence showing cost-effectiveness. Further studies on public perspectives towards vaccine distribution and the budget burden are needed. Last, as the time horizon of the analysis was quite short, we did not account for the discount rate for economic parameters. In addition, our analysis might not be valid (especially for long-term outcomes) if there is a remarkable change in the knowledge available. So far, knowledge on COVID-19 had been evolving in almost every aspect. It is also universally accepted that genetic mutations of SARS-CoV-2 occur over time, meaning that the VE of all vaccines is never constant. Regular updating on VE studies and other essential parameters related to COVID-19 is therefore indispensable.

## 5. Conclusions

The migrant-centric vaccination policy demonstrated the most effective outcome, both in terms of cost per life saved and cost per GDP recovery. This finding remained robust even when vaccine effectiveness was diminished. A likely explanation is that the migrant-centric vaccination strategy tackled the disease directly at the center of the population at high risk of transmission. Nonetheless, in the real-world implementation, other aspects of policy should be considered. These include studies on budget impact and societal attitudes towards the target recipients of the vaccines.

## Figures and Tables

**Figure 1 ijerph-18-10803-f001:**
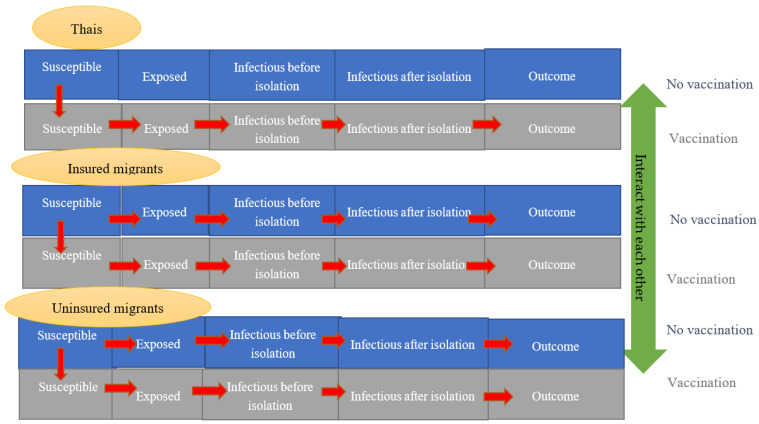
Model framework.

**Figure 2 ijerph-18-10803-f002:**
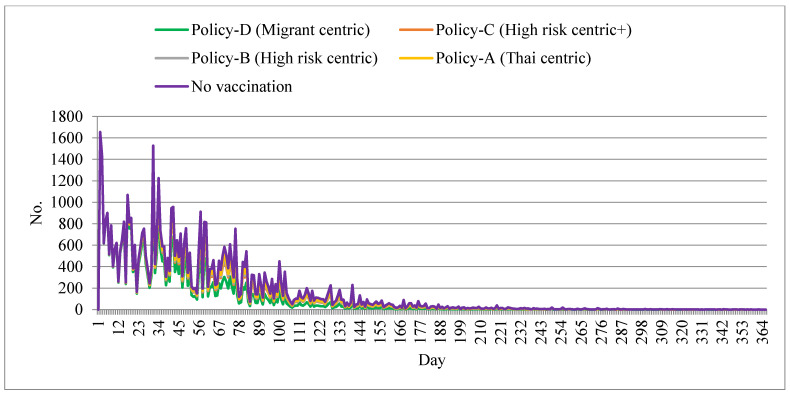
Daily new cases by different policy scenarios.

**Figure 3 ijerph-18-10803-f003:**
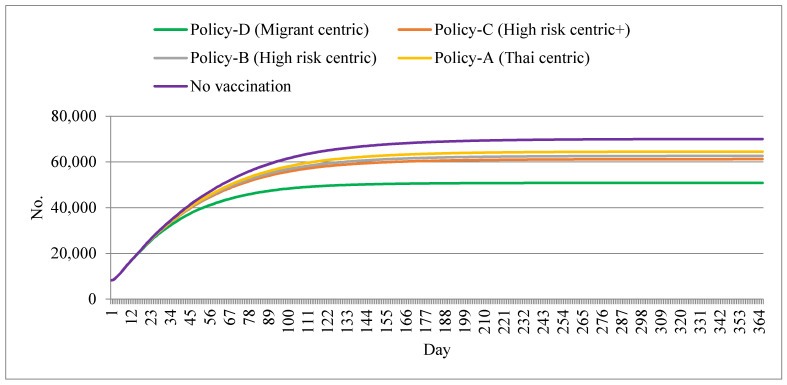
Cumulative cases by different policy scenarios.

**Figure 4 ijerph-18-10803-f004:**
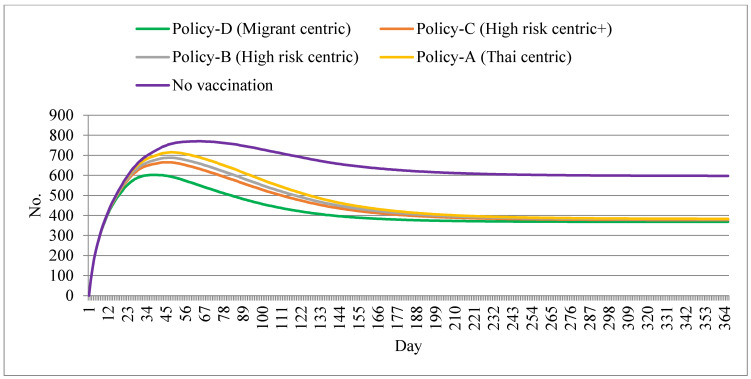
Prevalent ICU bed demand by different policy scenarios.

**Figure 5 ijerph-18-10803-f005:**
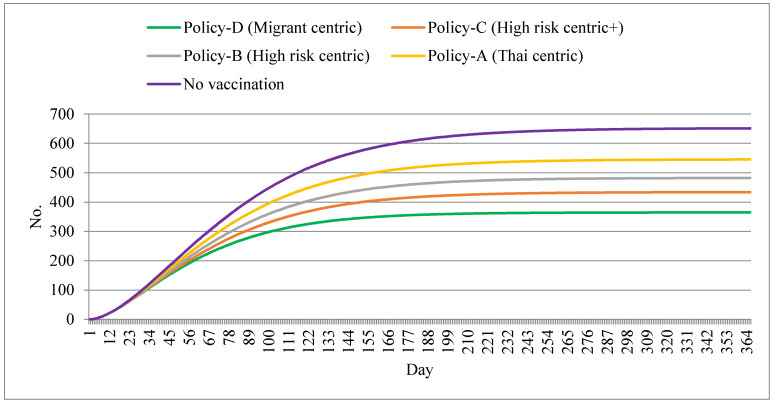
Cumulative deaths by different policy scenarios.

**Figure 6 ijerph-18-10803-f006:**
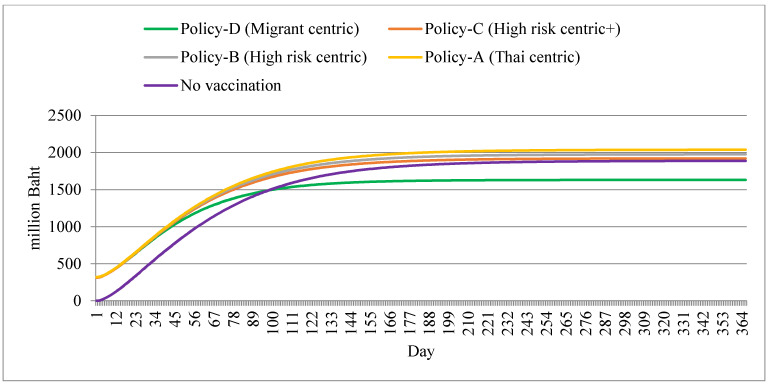
Total state-born cost by different policy scenarios.

**Figure 7 ijerph-18-10803-f007:**
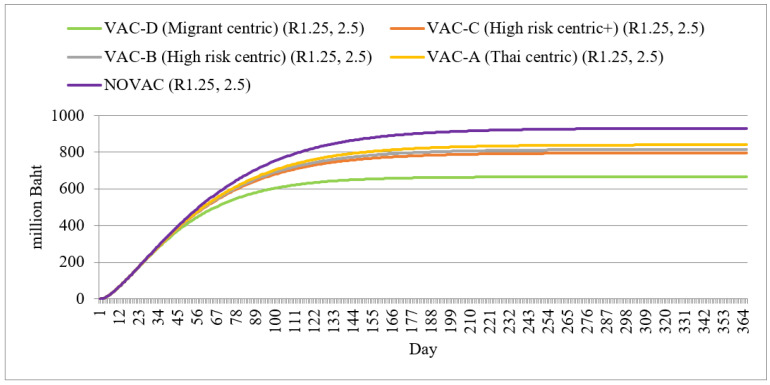
GDP loss by different policy scenarios.

**Table 1 ijerph-18-10803-t001:** Probability of contact for each population subgroup.

Contact Probability	Thai	Registered Migrants	Unregistered Migrants
Thai	60%	20%	20%
Registered migrants	20%	60%	20%
Unregistered migrants	20%	20%	60%

**Table 2 ijerph-18-10803-t002:** Risk distribution of the target populations.

Populations	Thais	Registered Migrants	Unregistered Migrants
Low risk (younger age and disease-free)	84%	94%	94%
High risk (aged over 60 years or with underlying diseases)	15%	5%	5%
Special risk (healthcare workers or living in epidemic centers)	1%	1%	1%

**Table 3 ijerph-18-10803-t003:** Policy options based on a total of 200,000 vaccinated persons.

Population	Policy A		Policy B		Policy C		Policy D	
	Percentage Target for Vaccination	Number	Percentage Target for Vaccination	Number	Percentage Target for Vaccination	Number	Percentage Target for Vaccination	Number
High-risk Thai	100%	75,000	100%	75,000	100%	75,000	100%	75,000
Low-risk Thai	29%	120,000	25%	106,500	24%	102,000	0%	0
Special-risk Thai	100%	5000	100%	5000	100%	5000	100%	5000
High-risk registered migrants	0%	0	100%	13,500	100%	13,500	100%	13,500
Low-risk registered migrants	0%	0	0%	0	0%	0	39%	99,300
Special-risk registered migrants	0%	0	0%	0	0%	0	100%	2700
High-risk unregistered migrants	0%	0	0%	0	100%	4500	100%	4500
Low-risk unregistered migrants	0%	0	0%	0	0%	0	0%	0
Special-risk unregistered migrants	0%	0	0%	0	0%	0	0%	0
Sum of vaccinated people	-	200,000	-	200,000	-	200,000	-	200,000

**Table 4 ijerph-18-10803-t004:** List of key parameters.

Parameters	Unit *	Value	Reference
Reproduction number among Thais	Dimensionless	1.25	Model calibration
Reproduction number among migrants	Dimensionless	2.5	Model calibration
Thai population in Samut Sakhon	Persons	500,000	NSO Thailand [15]
Registered migrants in Samut Sakhon	Persons	270,000	Department of Employment, MOL [16]
Unregistered migrants in Samut Sakhon	Persons	90,000	Calculated from the ratio of registered migrants to unregistered migrants [17]
Prevalence of infectees at the beginning of the outbreak among Thais (%)	Dimensionless	2	Expert opinions
Prevalence of infectees at the beginning of the outbreak among migrants (%)	Dimensionless	0.2	Expert opinions
VE against infection (%)	Dimensionless	50	WHO [14,20]
VE against severe illness and death (%)	Dimensionless	50	Model assumption [14,20]
Average infectious duration	Days	4.8	Ganyani et al. [22]
Average incubation period	Days	5	McAloon et al. [23]
Gap between infected and isolated	Days	3.5	Expert opinions
Clinical profile (low-risk or special-risk groups) given no vaccination: asymptomatic (%)	Dimensionless	60	Internal database of the DDC
Clinical profile (low-risk or special-risk groups) given no vaccination: mild-to-moderate (%)	Dimensionless	36	Internal database of the DDC
Clinical profile (low-risk or special-risk groups) given no vaccination: severe (%)	Dimensionless	3.5	Internal database of the DDC
Clinical profile (low-risk or special-risk groups) given no vaccination: dead (%)	Dimensionless	0.5	Internal database of the DDC
Clinical profile (high-risk group) given no vaccination: asymptomatic (%)	Dimensionless	53	Internal database of the DDC
Clinical profile (high-risk group) given no vaccination: mild-to-moderate (%)	Dimensionless	34.5	Internal database of the DDC
Clinical profile (high-risk group) given no vaccination: severe (%)	Dimensionless	5.5	Internal database of the DDC
Clinical profile (high-risk group) given no vaccination: dead (%)	Dimensionless	7	Internal database of the DDC
Clinical profile (low-risk or special-risk groups) given vaccination: asymptomatic (%)	Dimensionless	80.05	Internal database of the DDC and model assumption based on VE
Clinical profile (low-risk or special-risk groups) given vaccination: mild-to-moderate (%)	Dimensionless	18	Internal database of the DDC and model assumption based on VE
Clinical profile (low-risk or special-risk groups) given vaccination: severe (%)	Dimensionless	1.7	Internal database of the DDC and model assumption based on VE
Clinical profile (low-risk or special-risk groups) given vaccination: dead (%)	Dimensionless	0.25	Internal database of the DDC and model assumption based on VE
Clinical profile (high-risk group) given vaccination: asymptomatic (%)	Dimensionless	76.5	Internal database of the DDC and model assumption based on VE
Clinical profile (high-risk group) given vaccination: mild-to-moderate (%)	Dimensionless	17.25	Internal database of the DDC and model assumption based on VE
Clinical profile (high-risk group) given vaccination: severe (%)	Dimensionless	2.75	Internal database of the DDC and model assumption based on VE
Clinical profile (high-risk group) given vaccination: dead (%)	Dimensionless	3.5	Internal database of the DDC and model assumption based on VE
Gross domestic product of Samut Sakhon	Billion Baht	485	National Statistical Office [24]
Contribution of migrants to the provincial gross domestic product (%)	Dimensionless	20	Model assumption
Length of hospital stay (asymptomatic)	Days	10	Internal database of the DDC
Length of hospital stay (mild-to-moderate)	Days	14	Internal database of the DDC
Length of hospital stay (severe)	Days	21	Internal database of the DDC
Length of hospital stay (dead)	Days	21	Internal database of the DDC
Treatment unit cost per person per visit (asymptomatic)	Baht	13,265	Internal database of the NHSO
Treatment unit cost per person per visit (mild-to-moderate)	Baht	37,142	Internal database of the NHSO
Treatment unit cost per person per visit (severe)	Baht	113,172	Internal database of the NHSO
Treatment unit cost per person per visit (dead)	Baht	172,453	Internal database of the NHSO
Vaccination cost per person (vial)	Baht	1200	Terry [25]
Vaccine administration cost per person	Baht	400	Meeyai et al. [26] ^§^

Note: * US$1 = 31.9 Baht (as of 29 June 2021); ^§^ Adopted from the administration cost for flu vaccination in Thai late juveniles. Abbreviations: DDC—Department of Disease Control; MOL—Ministry of Labor; NHSO—National Health Security Office; NSO—National Statistical Office; VE—Vaccine Effectiveness; WHO—World Health Organization.

**Table 5 ijerph-18-10803-t005:** Essential formula of the model.

Change of Status	Formula	Note
From susceptible to exposed (amongst the vaccinees)	−(R/Dinf) × (1−VE)−S × Ib/P	Dinf = infectious duration, Ib = Infectees before isolation, P = total population, R = basic reproduction number, S = susceptible population, VE = vaccine effectiveness
From susceptible to exposed (amongst the non-vaccinees)	−(R/Dinf) × S × Ib/P	Dinf = infectious duration, Ib = Infectees before isolation, P = total population, R = basic reproduction number, S = susceptible population, VE = vaccine effectiveness
From susceptible to infectious (before isolation)	−E/Dinc	Dinc = incubation period, E = Exposed population
From infectious (before isolation) to infectious (after isolation)	−Ib/Dlag	Dlag = lag days from infection to isolation, Ib = Infectees (before isolation)
From infectious to recovered	−Ia/Drx	Drx = hospitalization days, Ia = Infectees (after isolation)

**Table 6 ijerph-18-10803-t006:** Cost-effectiveness analysis by different policy scenarios.

Comparison of Policy Options	Incremental Cost	Cases Averted	GDP Resumed (m. Baht)	Incremental Cost Per One Case Averted	Incremental Cost Per One Life Saved	Incremental Cost Per One Baht GDP Resumed
A vs. no- vaccination	150,105,789	5520	90.2	27,191	1,421,336	1.7
B vs. no- vaccination	87,966,921	7409	115.1	11,873	521,613	0.8
C vs. no- vaccination	33,394,776	8829	134.3	3782	153,940	0.2
D vs. no- vaccination	−256,956,014	19,200	264.4	−13,383	−897,755	−1.0

**Table 7 ijerph-18-10803-t007:** Cost-effectiveness analysis by varying reproduction numbers.

Policy Options; Fixing VE at 50%	R = (1.25, 3)		R = (1.25, 2.5)		R = (1.25, 2)		R = (1.25, 1.5)	
	Incremental Cost Per One Life Saved	Incremental Cost Per One Baht GDP Resumed	Incremental Cost Per One Life Saved	Incremental Cost Per One Baht GDP Resumed	Incremental Cost Per One Life Saved	Incremental Cost Per One Baht GDP Resumed	Incremental Cost Per One Life Saved	Incremental Cost Per One Baht GDP Resumed
A vs. no-vaccination	28,633	0.03	1,421,336	1.7	6,484,316	8.7	16,196,527	26.2
B vs. no-vaccination	−394,666	−0.6	521,613	0.8	4,099,922	7	11,649,945	23.8
C vs. no-vaccination	−512,873	−0.9	153,940	0.2	2,934,559	5.6	9,114,107	21
D vs. no-vaccination	−1,553,363	−1.6	−897,755	−1	1,656,283	2.2	7,604,773	14.2

Note: R (X, Y) = (R among Thais, R among migrants); VE = vaccine effectiveness.

**Table 8 ijerph-18-10803-t008:** Cost-effectiveness analysis by varying vaccine effectiveness.

Policy Options; Fixing R = (1.25, 2.5)	VE = 0.2		VE = 0.5		VE = 0.8	
	Incremental Cost Per One Life Saved	Incremental Cost Per One Baht GDP Resumed	Incremental Cost Per One Life Saved	Incremental Cost Per One Baht GDP Resumed	Incremental Cost Per One Life Saved	Incremental Cost Per One Baht GDP Resumed
A vs. no-vaccination	3,631,683	7.4	1,421,336	1.7	502,188	0.5
B vs. no-vaccination	1,894,715	5.2	521,613	0.8	−98,760	−0.1
C vs. no-vaccination	1,234,785	3.8	153,940	0.2	−355,361	−0.5
D vs. no-vaccination	222,256	0.4	−897,755	−1.0	−1,290,845	−1.2

Note: R (X, Y) = (R among Thais, R among migrants); VE = vaccine effectiveness.

## Data Availability

The datasets generated and/or analyzed during the current study are not publicly available due to the Thai-DDC’s regulation but are available from the corresponding author on reasonable request.

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
