# Peer review of "Prioritization of the Target Population for Coronavirus Disease 2019 (COVID-19) Vaccination Program in Thailand"

_ijerph, 2021, doi:10.3390/ijerph182010803_

Round 1

Reviewer 1 Report

Thanks for submitting your manuscript on a timely issue. It reads well but there are some areas for improvement mentioned below-

  1. Please include the article in the introduction of the manuscript if possible -https://doi.org/10.3390/healthcare9020156
  2. Figure 1 seems like a screenshot, please provide a high quality image
  3. In the discussion, use the following article for compare the findings - 

https://doi.org/10.3390/vaccines9080864

4. In the methods, just mention secondary data analysis.

5. The conclusion does not reflect the findings, please modify it

Author Response

 Please find attached the new Word document.

Reviewer 2 Report

This is an excellent paper with very timely and important findings.

Some notes for clarification:

1) Figure 2 is showing some stochastic behaviour, this is interesting for deterministic model. Is the incidence stochastic , how are multiple peaks appearing? do you change the parameters for each day?

2) Table 3. should read 200,000 vaccinated people.

3) Is your STELLA model available to other researchers?

Author Response

Dear editors and reviewers

Thank you so much for the comments on the manuscript. We do appreciate your help in making this manuscript to have a better shape. We have addressed the comments on a point-by-point basis. Please find our responses below. The marked-up manuscript and the clean manuscript are re-submitted in parallel with this letter.

Best regards

All authors

Reviewer 3 Report

This is an interesting and worthwhile paper on an important topic in the prevention of COVID-19 worldwide.
1. In doing this, they make some assumptions about the reader's understanding of their methodology that makes the paper more difficult to read. The paper can be improved by being more explicit about how they estimated the different frames, and the underlying assumptions.
2. It is also not very clear how values for the non-vaccinated group were determined, was there any empirical data from this group? If not, then the assumptions about this group of patients underlying these estimates should also be made clear.
3. Vaccination coverage and attendance is also an important factor in a prevention programme, and the authors should make some consideration about those who were lost to follow up or simply did not attend for screening, and how this would affect the resources and success the recommended vaccination.
4. Did the authors take into account the indirect costs of complications from a societal perspective? If not, then the perspective of the present analysis should also be made more clear.
5. How is the discount rate? It would be helpful for other researchers looking at the same types of issues to replicate this approach to know how to appropriately discount for their population.
6. More discussion regarding the policy implications of their findings would be important for the use of the methodology in health policymaking.
7. The English should be improved.

Author Response

(The authors gave the same response as above.)

Reviewer 4 Report

This paper documented that beneficial effect of targeting vaccination to migrant population from the viewpoint of public-health outcomes and economy. It was well written and will be an interest to the readers.

Minor comments:

Line 355; "some some" may be a mistake. 

Author Response

(The authors gave the same response as above.)
